# Development of a City-Scale Approach for Façade Color Measurement with Building Functional Classification Using Deep Learning and Street View Images

**Jiaxin Zhang** , **Tomohiro Fukuda \*** and **Nobuyoshi Yabuki**

Division of Sustainable Energy and Environmental Engineering, Graduate School of Engineering,
Osaka University, 2-1 Yamadaoka, Suita, Osaka 565-0871, Japan; zhang@it.see.eng.osaka-u.ac.jp (J.Z.);
yabuki@see.eng.osaka-u.ac.jp (N.Y.)
\* Correspondence: fukuda@see.eng.osaka-u.ac.jp

**Abstract:** Precise measuring of urban façade color is necessary for urban color planning. The existing manual methods of measuring building façade color are limited by time and labor costs and hardly carried out on a city scale. These methods also make it challenging to identify the role of the building function in controlling and guiding urban color planning. This paper explores a city-scale approach to façade color measurement with building functional classification using state-of-the-art deep learning techniques and street view images. Firstly, we used semantic segmentation to extract building façades and conducted the color calibration of the photos for pre-processing the collected street view images. Then, we proposed a color chart-based façade color measurement method and a multi-label deep learning-based building classification method. Next, the field survey data were used as the ground truth to verify the accuracy of the façade color measurement and building function classification. Finally, we applied our approach to generate façade color distribution maps with the building classification for three metropolises in China, and the results proved the transferability and effectiveness of the scheme. The proposed approach can provide city managers with an overall perception of urban façade color and building function across city-scale areas in a cost-efficient way, contributing to data-driven decision making for urban analytics and planning.

**Keywords:** façade color measurement; building classification; street view images; deep learning; urban analytics; urban computing

## 1. Introduction

In the last several decades, empirical observations and scientific studies have proven that human-environment reaction in an urban environment is primarily based on the sensory perception of its color [1]. Therefore, city managers have attached great importance to urban color and issued a series of color planning rules in urban development [2]. Urban color planning can guide the use of color to achieve color harmonies in an urban environment [3]. The urban function is one of the critical factors that designers need to consider when implementing urban color planning [2]. Many urban planners have suggested that buildings for specific functions need to comply with a spectrum of colors [4]. However, the acceleration of urbanization poses troubles for urban color planning [5]. Some urban designers do not consider the influence of the surrounding color of the environment when developing new buildings or renovating existing ones [6]. Many emerging architectural styles are becoming similar in fast-growing cities, resulting in the loss of urban color identity [7].

Fine-grained façade color measurement with building functional classification at a large scale has become an essential basis for urban color planning and data-driven city management. The general procedure of previous field survey-based methods consisted of the following steps: (1) data collection in the target area through building photo shooting

and building function auditing; (2) identification of the façade dominant color and land use in the target area; and (3) developing urban color guidelines based on principles such as functional matching and color coordination [8]. For instance, Li et al. analyzed the association between different urban functional areas and the urban façade colors in Luoyang city through field research and identified the spatially sensitive areas of urban colors based on city image theory [9]. Nguyen et al. conducted manual measurement to characterize chromatic attributes of several functional areas, allowing designers to determine the coordination ranks of urban colors [10].

However, the conventional methods rely heavily on extensive field survey data and site-specific analysis, which is labor-intensive and unsustainable [11]. Manual measurements are effective at the neighborhood level but challenging to adapt to the macro-scale. Moreover, they cannot accommodate data updates in fast-growing areas or provide sufficient city-scale data for fine-grained urban management. Recently, thanks to rapid advances in city databases and computer vision techniques, we can use urban public data and deep learning algorithms to perform an in-depth analysis of the built environment. We can use semantic segmentation to extract buildings from street-level images and calculate the façade color [12]. In addition, the calibration methods of digital photos can be used to correct variation in saturation and brightness for improving the accuracy of color measurements [13]. The building function can be automatically classified from street view images using image classification techniques [14]. In summary, these emerging technologies show great promise to effectively extract helpful information from street view imagery and potentially support façade color measurement with building classification at a city scale.

The main objective of this study is to develop a quantitative analysis method for façade color measurement and building functional classification from the city scale. In the experiment, we select field samples from street-level images to verify the accuracy of our approach and adopt three cities in China for the mapping of urban façade colors and building functions. In general, the main contributions are listed as follows:

- Compared with methods through field survey measurements, this paper developed an automatic method for façade dominant color measurement with building functional classification using state-of-the-art deep learning models and extensive-coverage street view images, significantly improving the efficiency of city-scale data analysis.
- We applied our method to three metropolises of China. The validation results demonstrated that our approach is generalizable with satisfactory accuracy in building segmentation, façade dominant color calculation, and classifying building functions.
- A tailored street-view dataset was built for training multi-label classifiers of building functions, including residential, public services, commercial services, and other facilities.

The rest of this paper is organized as follows: The relevant literature review is briefly introduced in Section 2. Section 3 presents the research methodology, elaborating on the research framework and the technical process of extracting building colors and classifying building functions from large-scale street view images. Section 4 validates the effectiveness of the proposed method and presents the mapping results of façade color and building function classification for three cities in China. Section 5 discusses the advantages, potential applications, and limitations of the proposed method. Section 6 provides the conclusions of the study.

## 2. Literature Review

Academic discussions and applications of city-scale measurement for urban façade color with building classification have revolved around four aspects.

### 2.1. Urban Color Planning Based on Function Classification

As early urban and architectural designers believed in the principle of form following function, functionality classification plays a crucial role in urban color planning [15]. In general practice, there were two methods of functional-based color planning, namely the

color spectrum method and the primary tone method. The former emphasizes the zoning control of colors [2], and the latter method extracts several colors as the primary color for urban color planning [16]. These two methods were human interventions in color identity based on functional rationalism, leading to the close relationship between urban color and urban function. Most urban color plans considered controlling the building color to be critical because buildings were more extensive and complex than other products created by human beings. However, in many practical cases, the color design of many buildings had no logic and was improvised by the designers, causing the disappearance or homogenization of urban color characteristics.

### 2.2. Façade Color Measurement

In the early phases of architectural and urban design, the façade color measurement was an essential data collection work that could be carried out on a large scale under a quantitative definition of the color system. The Munsell color system is commonly used in urban color management to holistically and intuitively perceive the change of color-related characteristics [17]. The constant color was one of the most complicated functions of the human visual system because ambient light has a significant influence on the color stimulus, and architectural colors in photos often have chromatic aberrations. Therefore, the collected architectural photos should be color-corrected before statistics and analysis. Automatic white balance (AWB) and automatic exposure correction (AEC) are image processing steps in the digital camera imaging pipeline, calibrating the façade color in photos to ensure that the data have high fidelity [18,19]. The manual color measurement was the primary survey method in urban color management. However, as entering the post-urban development era, the demand for intelligent management of cities has increased. Previous methods on small samples were unable to find a connection between theoretical study and practical operation. They deepened their logical reasoning through induction and deduction rather than exploring based on the principles of large-scale investigations.

### 2.3. Functional Classification of Buildings

The functional classification of buildings typically includes six categories based on building land-use in the urban area, namely residence (R), public service (A), commercial service (B), transportation facilities (S), greening (G), and utilities (U) [20]. Many automatic building classification methods have emerged with the development of measurement tools and data sources [21]. The classification methods combined with street view images and deep learning are open source and favored by city researchers in urban analytics and urban computing, showing great potential for many applications, such as urban population mapping [22], density analysis [23], or urban utility planning [24]. However, there are usually multiple buildings in street view photos, and it is necessary to move beyond the single-label classification tasks to precisely describe the building classification in the image. Since the input image and output label spaces have various types and quantities, multi-label classification can describe more information than single-label classification [25]. The multi-object classifier is competent for classifying buildings in high-functional mixed areas of the central city.

### 2.4. Problem Statements of This Research

According to the above literature review, the main issues anticipated for this study include street view data acquisition and cleaning, building façade extraction, façade color calculation, and building function classification. To achieve our research objectives and bridge the gap, the following challenges are worth noting. First, for data acquisition and cleaning, many studies measuring physical elements at large scales from street-level images have achieved great success, such as urban canyon classification [26], green view segmentation [27], and detection of building façades [28]. However, weather and sunlight can have an impact on the quality of street view images, affecting the accuracy of color-based calculation and image classification. Second, for deep learning-based façade segmentation,

many datasets containing building categories have been established, such as Cityscape [29] and PASCAL VOC [30] datasets, but the accuracy of the pre-trained building segmentation model on these datasets in different cities needs further validation. Third, some dominant color description quantification methods have been proposed, such as standard color chart-based [8] and histogram-based approaches [31]. It is necessary to calculate the façade dominant colors from photos on the standard architectural colors, which can facilitate alignment with urban color planning. Finally, for deep learning-based building functional classification, the accuracy of deep convolutional neural networks (DCNNs) varies greatly in the same task, and it is important to compare the state-of-the-art DCNNs and perform urban computations using a high-accuracy model.

### 3. Methods

The workflow of the presented method is shown in Figure 1. To develop a system for automatic calculation and identification of urban façade colors and functions at the city scale, we first collect a large number of street view pictures as data support. Then, we apply the color calibration and data cleaning methods for street-level images. Finally, we present a color chart-based façade color measurement and a multi-label deep learning network for building classification.

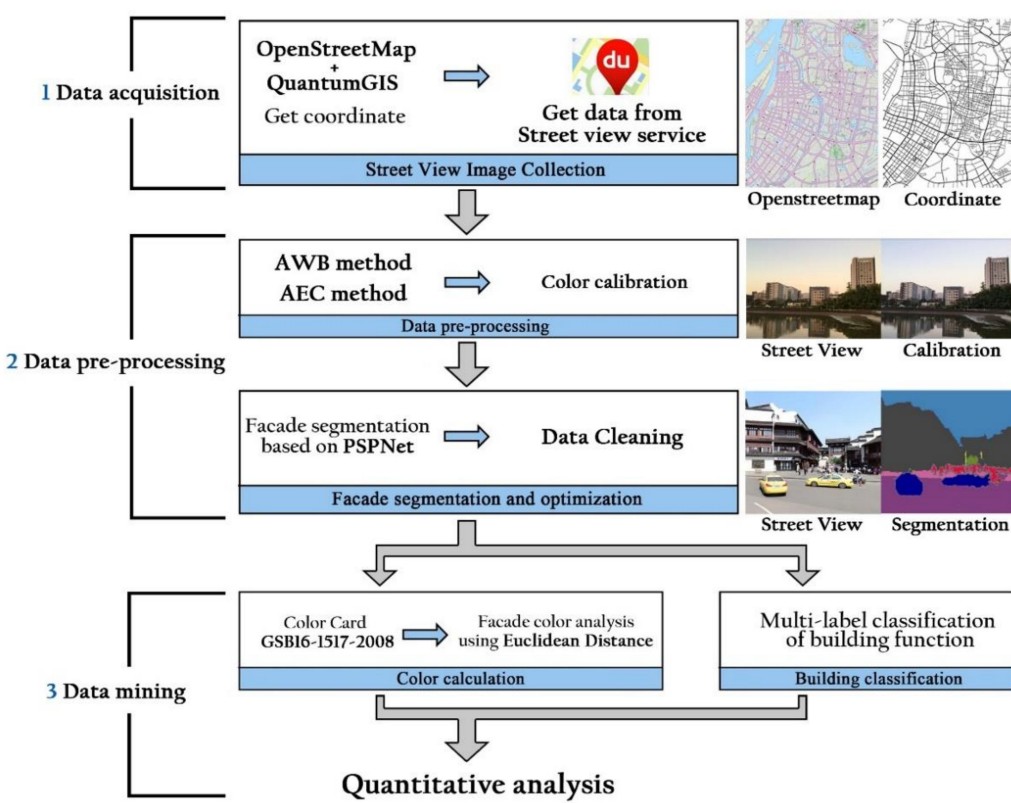

**Figure 1.** Workflow for façade color measurement with building classification.

### 3.1. Data Acquisition

Pictures can be extracted from the street view platform to provide street view photos with extensive coverage in an urban street. Firstly, urban road networks with geography coordinate information were selected and obtained from OpenStreetMap (OSM) [32]. Then, the road networks were simplified into single lines with an average distance of 20 m between adjacent points adopted from the urban street design methodology of J. Gehl [33]. Next, the sampling points with geographical coordinate information can be obtained and shown in spatial distribution. However, it is worth noting that not all sampled points in the street view service have corresponding street view images. Lastly, to obtain the building façade, we downloaded two pictures (including left and right) perpendicular to

the road from the street view service (the viewing angle is 90 degrees, the horizontal angle is 0 degrees, image size is 800 × 500 pixels) for each sampling point (as shown in Figure 2).

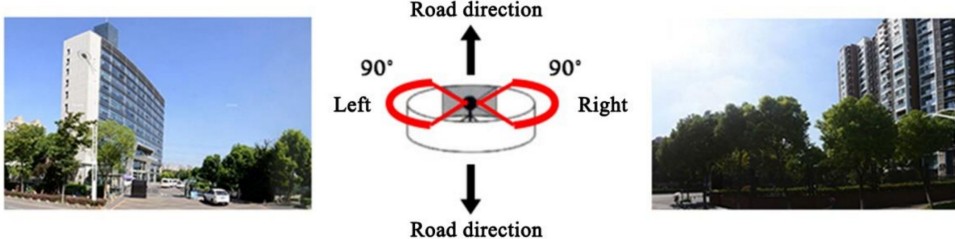

**Figure 2.** Street-level imagery acquisition at the sampling point.

### 3.2. Data Pre-Processing

### 3.2.1. Color Calibration of Street View Images

The ambient light has a significant effect on the color stimulus. If the color temperature of the sunlight is cold, the object being captured will appear bluish. On the other hand, the object will appear reddish with a warm temperature light source [34]. Since the saturation and brightness of street view images are affected by weather and time, eliminating the deviation affected by ambient light is the analysis premise. The previous study has shown that hue, saturation, and value (HSV) color space has better color calibration performance than red, green, and blue (RGB) channels [35]. Therefore, we converted the collected images to HSV color space. The AWB method was used for the saturation calibration of the street view images (for the basic method principles, referring to Lam et al. [18]). In addition, the AEC of the digital photographs method proposed by Yuan et al. [19] was introduced to adjust overexposed and overly dark street view images. Figure 3 shows the calibration demo by AWB and AEC.

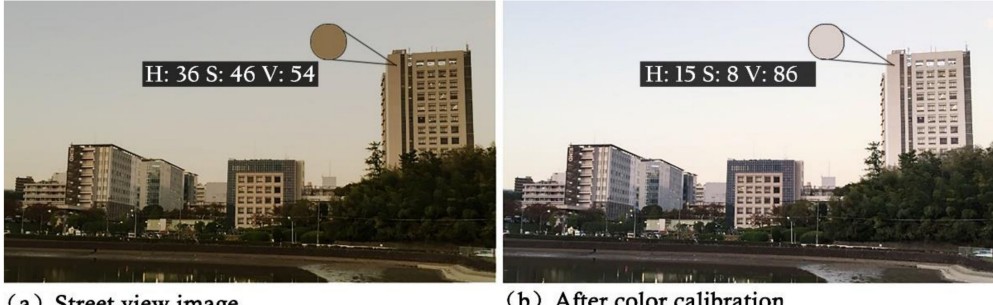

**Figure 3.** A color calibration demo by AWB and AEC methods. (**a**) original street view image and (**b**) after color calibration.

### 3.2.2. Building Façade Segmentation and Data Cleaning

The building façade should be segmented from the street view image to improve the detection accuracy of identifying the façade color and classifying the building functions. Various convolutional network-based semantic segmentation models with high accuracy have been developed recently, such as U-Net [36], DeepLabv3 [37], and PSPNet [38]. In this study, the Pyramid Scene Parsing Network (PSPNet) was used to segment the building façades from street view images because it is highly accurate and easily accessible. Unlike the other methods using RGB values to extract building elements, the network structure of PSPNet has been widely used, where spatial statistics provide a good descriptor for explaining the overall scene. The single PSPNet yields a record of mIoU accuracy of 85.4% on PASCAL VOC 2012 and 80.2% accuracy on Cityscapes [38]. Our study used the Cityscapes dataset as the training data, and IoU accuracy of building on the Cityscapes test set is 92.6%. Figure 4 shows the building segmentation results of street view images

by the proposed trained PSPNet. However, there is a low proportion of buildings in some street view images, and these building images cannot reflect the color characteristics of the building. To improve the experiment's accuracy, we needed to delete the pictures with a small proportion of buildings because the computer cannot recognize the features of the buildings through these images. The ratio of the building façade area can be measured by inputting street-level images into the pre-trained semantic segmentation model to generate segmentation results. By calculating the building proportion for each sampled image, we removed pictures with less than 20% building proportion as example Figure 4c shows.

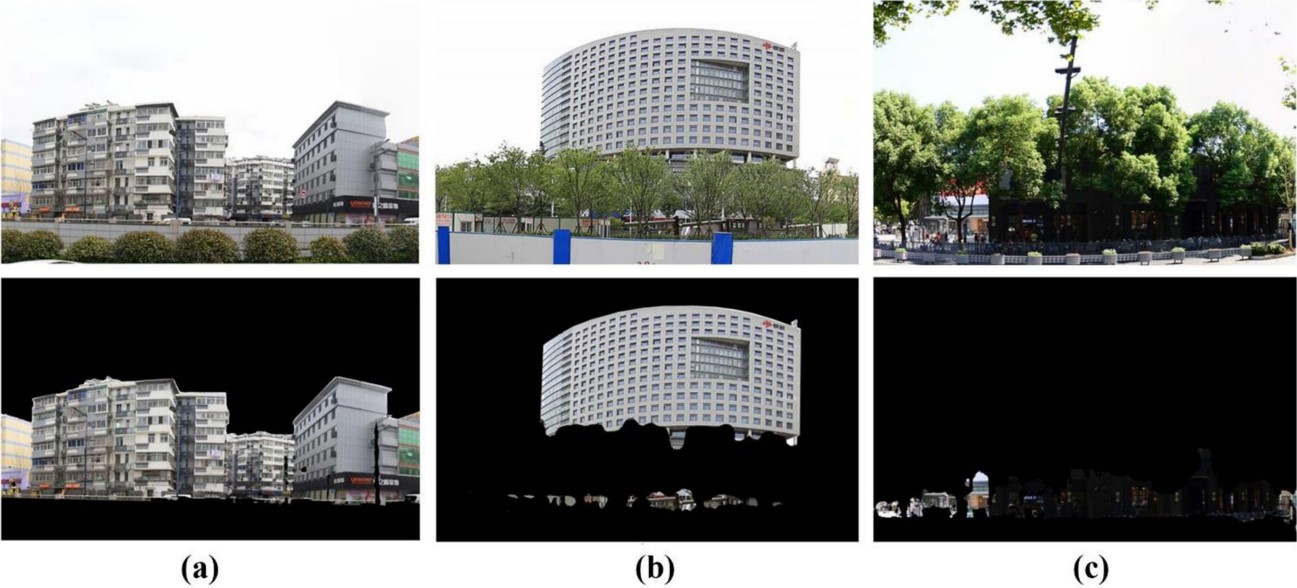

**Figure 4.** Street view image segmentation through Pyramid Scene Parsing Network. (**a**) Unobstructed façades, (**b**) obscured façades, (**c**) a low building proportion.

### 3.3. Data Mining

#### 3.3.1. Dominant Façade Color Calculation

There are thousands of color values in an image, and it is difficult to define the dominant color without merging colors. Therefore, extracting the dominant color of the urban façade requires a standard color card for integrating the colors in the image to the standard color. Since the use of color in architectural design and building decoration should conform to standard color codes in different countries, this study chose China Building Color Chart (CBCC)-258 as the standard color (the CBCC-258 selects 258 commonly used architectural colors from the complete CBCC library), which can cover most building colors in urban façades. The specific HSV information of CBCC-258 can be found in the online color chart [39]. Then, we merged the raw color data of the street view images with the standard color chart by calculating the HSV value of the street view color and replacing them with the closest architectural standard color (in terms of the Euclidean distance). In the HSV color space model, we were able to define the three-dimensional coordinate $(x, y, z)$ of the color point (H, S, V) according to Equation (1):

$$\begin{cases} x = r \cdot v \cdot s \cos h \\ y = r \cdot v \cdot s \sin h \\ z = L(1 - v) \end{cases} \tag{1}$$

where $r$ is the radius of the bottom circle, and $L$ is the height, and we take $r$ and $L$ to the integer 100 for the convenience of later analysis. $(h, s, v)$ is the HSV value of the image color. After calculating and merging the distance to the standard color, all colors on the street view images will be converted to the architectural standard color chart. Then, we can

count the color proportion from each street view picture. Although color dominance can be established in several aspects, such as the strength of hue, the sharpness of vision, contrast, and perception of saturation, G. A. Agoston suggested that the two most critical factors affecting the dominant color of the picture are the color proportion and the saturation contrast [40]. Therefore, the following is the approach of dominant color selection in this study:

- The dominant color should be the largest part of the building façade.
- When the color proportions are equal in a street view picture, the color with the highest saturation is the dominant color.

### 3.3.2. Multi-Label Classification of Building Function

From the perspective of the façade in urban streets, there are four main types of building functions in the city proper, namely residence (R), commercial service (B), public service (A), and other facilities (O) [41]. To effectively classify the types of buildings, we used a deep learning method to automatically identify the building functions in the street view images of the study areas. In the previous research, single-label methods have typically been used to classify building classes, with each photo corresponding to only one label [14]. However, the single-label method cannot accurately separate the street view pictures of several building functions, resulting in inaccurate experimental results. To solve this problem, we used a multi-label image classification method to identify multiple building categories in the street view images.

To train the multi-label building classifier, we first used the semantically segmented building images to build the corresponding street-view benchmark dataset that contains 4965 images from 4 basic categories: residential, commercial services, public services, and other facilities. Meanwhile, images with more than one label were classified as mixed services. The ground-truth labels of the training data are from the OSM, and Table 1 contains descriptions of the different building function classes. There are around 3500 single-label images and 1500 multi-label images in these training images, as shown in Figures 5 and 6. We divided these street-level images into a training set (75%) and a testing set (25%). It is worth noting that all test images are not retrieved from a single city and are different from those utilized for training. To augment the training data, we randomly selected $720 \times 450$ pixels from the original $800 \times 500$ pixels and flipped the cropped images horizontally. Then, we trained several state-of-the-art CNN-based models, including DenseNet [42], EfficientNet [43], InceptionNet_v4 [44], and ResNeSt [45], and demonstrated the corresponding classification performances. To improve the learning rate, we trained these models for 100 epochs and decayed the learning rate by a factor of 0.1 every 25 epochs. Each training batch contained a total of 64 images. Other not mentioned values were default. The experiments were implemented with Pytorch and conducted using one NVIDIA GeForce GTX 1080 Ti 11 GB GPU.

**Table 1.** Description of building class in the city.

| Building Classifications | Description |
|---|---|
| Residential (R) | Buildings are for people living, including villas, apartments, and dormitories. |
| Commercial service (B) | Buildings allow people to engage in various business activities, including retail, shopping malls, markets, hotels, restaurants, and entertainment facilities. |
| Public services (A) | Buildings allow people to carry out various public activities, including office, education, health, culture, transportation, and tourism buildings. |
| Other facilities (O) | Buildings or structures that appear in urban areas other than the above three. |

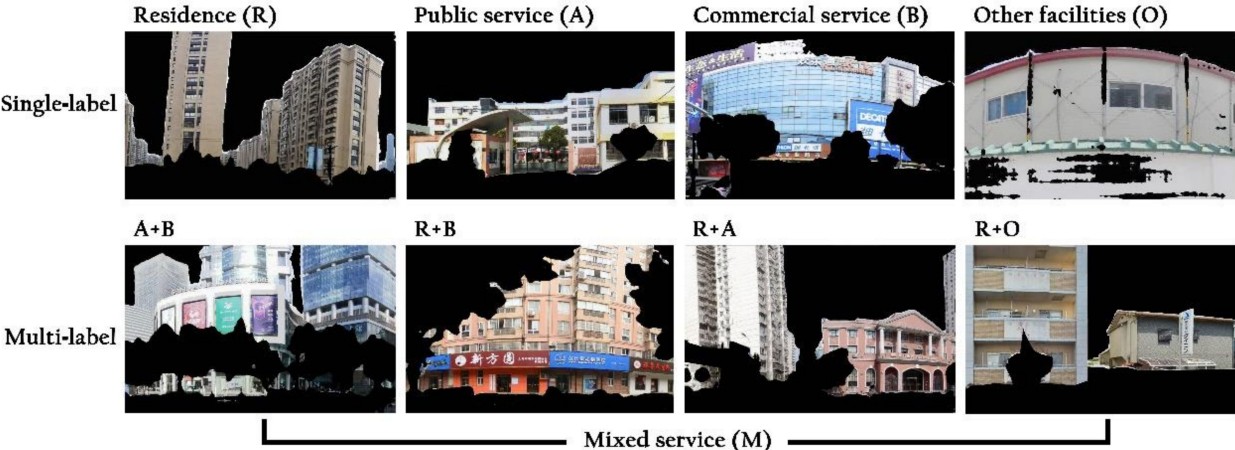

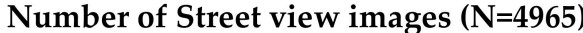

**Figure 5.** A total of 4965 street view images with four single-labeled and rearranged buildings are included. The first line is single-label class, from left to right: residential, public services, commercial services, and other facilities. The second line is the multi-label class, from left to right: public service and commerce, residence and commercial service, residence and public service, residence, and other facilities.

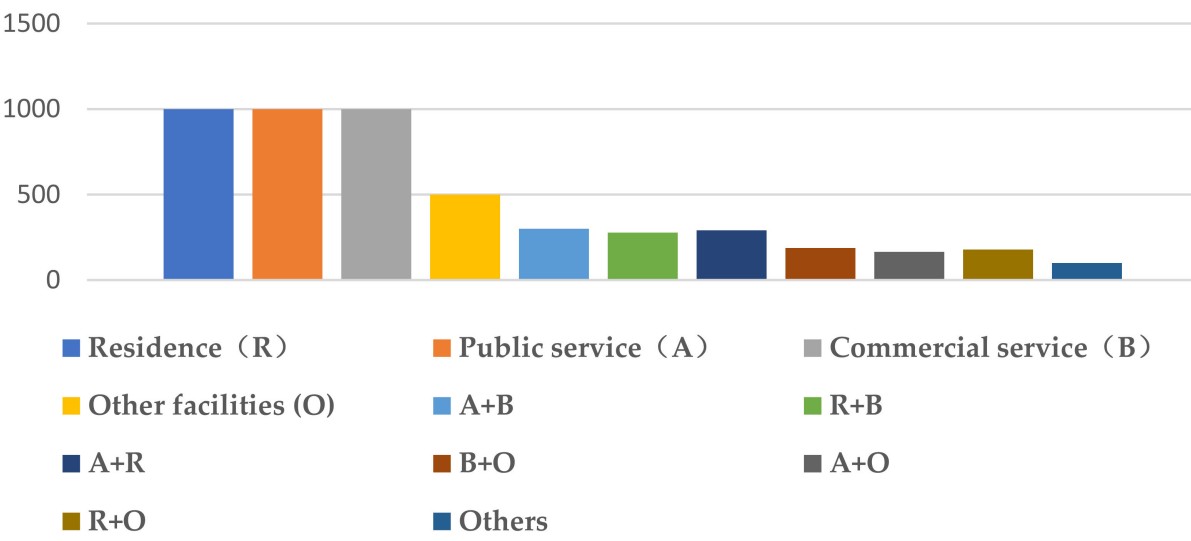

**Figure 6.** The number of training set images for each building category.

## 4. Experiment

### 4.1. Study Area

Our research is conducted within three metropolises, Shanghai, Nanjing, and Hefei, located in the Yangtze River Delta. In Chinese history, the Yangtze River Delta has long been a major center of economy, culture, education, politics, transport networks, and tourism; it is a multi-functional region with diverse city identities [46]. As shown in Figure 7, we choose the central areas of these three cities as the study area. For one, these regions are well covered by public street view services. For another, they retain the typical characteristics of these cities while freeing us from processing the entire city, which is impractically expensive. Specifically, in the study, Shanghai is approximately 124.6 square kilometers, Nanjing is approximately 152.1 square kilometers, and Hefei is approximately 108.6 square kilometers.

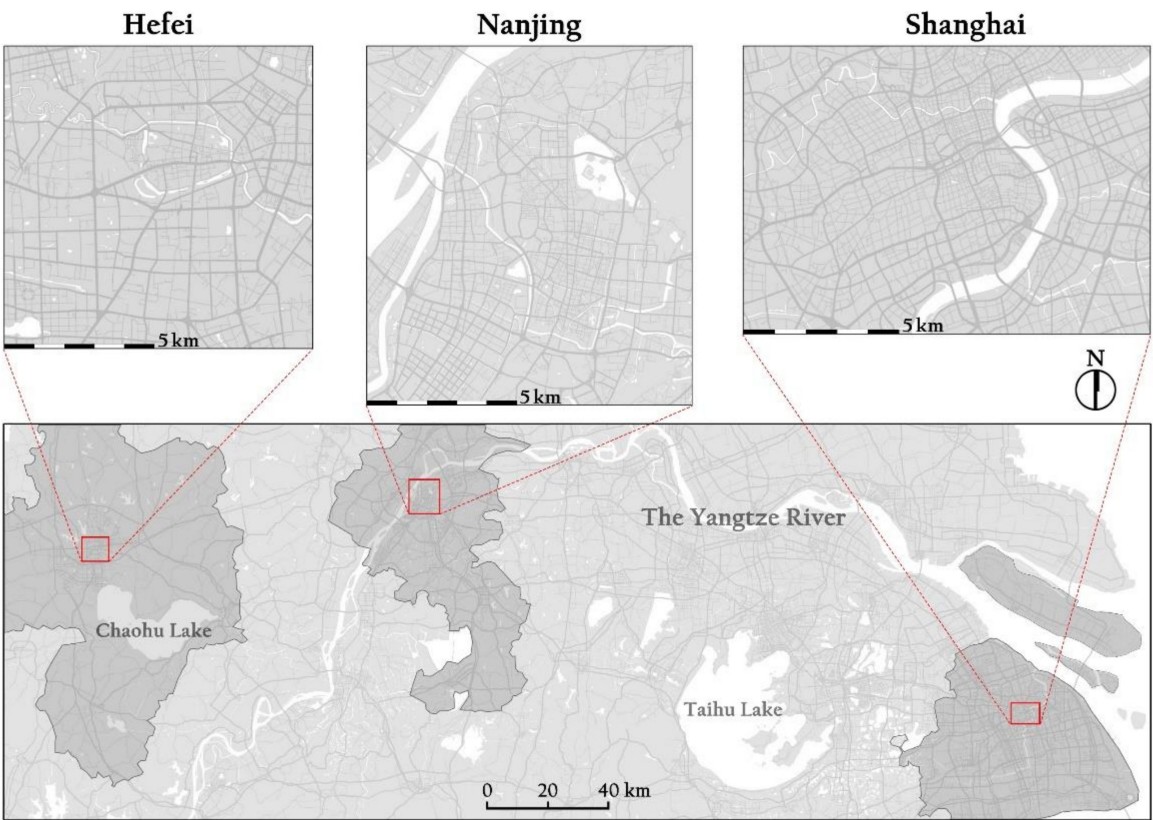

**Figure 7.** The study areas of the three metropolises (Shanghai, Nanjing and Hefei) in the Yangtze River Delta.

### 4.2. Valid Data for Experiments

To accurately carry out the color measurement and classification of the façade, we need to clean the collected street view data. It is worth noting that several cases in street view images are classified as invalid data, including the absence of buildings, buildings obscured by a large proportion of greenery, and street view images with severe color deviations. Table 2 gives detailed statistical results for valid data, including the street view images of Hefei, Nanjing, and Shanghai.

**Table 2.** Valid street view image statistics.

| City | Hefei | Ratio | Nanjing | Ratio | Shanghai | Ratio |
|---|---|---|---|---|---|---|
| Sampling points | 49,140 | | 59,420 | | 55,998 | |
| Total images | 94,244 | | 115,244 | | 110,974 | |
| Valid images | 74,760 | 79.3% | 79,204 | 69.7% | 102,046 | 88.5% |
| Invalid images | 19,484 | 20.7% | 36,040 | 31.3% | 13,198 | 11.5% |

### 4.3. Experimental Results

4.3.1. Accuracy Verification of Building Façade Segmentation

In this study, the building façade segmentation images were generated by PSPNet, which was pre-trained on the Cityscapes dataset. Since the accuracy of façade color is strongly influenced by the segmentation result, it is vital to verify the generalizability of the pre-trained model to the street view images of our study areas. To this end, we randomly selected 200 street-level photos (800 × 500 pixels resolution) with buildings from three cities and manually labeled the ground truth. Figure 8 shows an example of building

façade segmentation. To assess the segmentation performance, we measured the precision, recall, and intersection over union (IoU) of these 200 images as follows:

$$precision = \frac{TP}{TP + FP} \tag{2}$$

$$recall = \frac{TP}{TP + FN} \tag{3}$$

$$IoU = \frac{TP}{TP + FP + FN} \tag{4}$$

where for a given category, *TP* (true positive) represents the number of pixels that are correctly classified, *FP* (false positive) is the number of pixels that are incorrectly classified as belonging to this category, and *FN* (false negative) represents the number of pixels that are not classified as belonging to this category but should have been. As shown in Table 3, we achieved an IoU of 87.13% on average over 200 street-level images. This result shows the effectiveness of the pre-trained model on the extract buildings from the street views in our study areas, which is important for the subsequent façade color calculation.

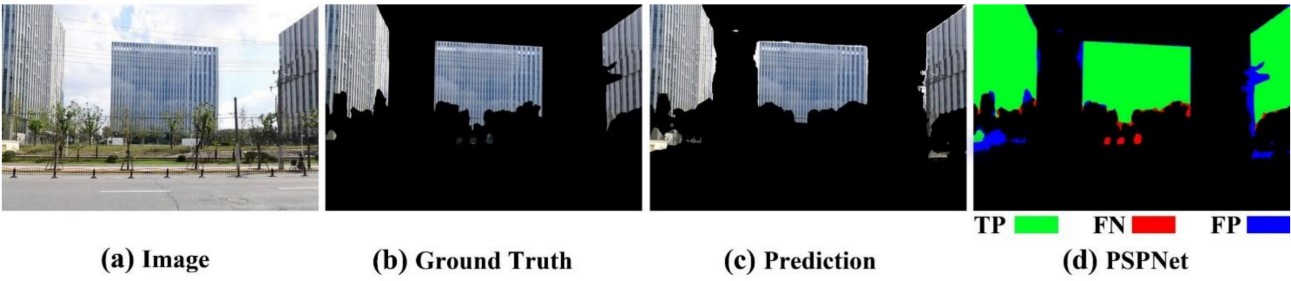

**(a) Image**  **(b) Ground Truth**  **(c) Prediction**  **(d) PSPNet**

**Figure 8.** A qualitative example of building façade segmentation by the pre-trained PSPNet model. (**a**) Original street view image. (**b**) Ground truth. (**c**) Building semantic segmentation by PSPNet. (**d**) Qualitative example, blue is true positive (TP), red is false negative (FN), and blue is false positive (FP).

**Table 3.** Building façade segmentation performance on street-level images.

| Model | Precision | Recall | IoU |
|---|---|---|---|
| PSPNet (%) | 92.07 | 93.30 | 87.13 |

4.3.2. Accuracy Verification of Façade Color Calibration

We first selected two materials (MAT. 1 is ceramic tiles, and MAT. 2 is veneer brick) with standard HSV information. Then, we used a digital camera to take ortho-projected photographs of the materials at six ambient color temperatures. Next, the AWB and AEC methods were used to conduct color calibration of the photos, and the corrected HSV values of the two materials can be obtained. Table 4 lists sample materials, the digital camera specifications, and the software used for the experiments. Finally, the shortest Euclidean distance between the standard HSV color value and the image color can be used to calculate the color deviation $\Delta E$, and Equation (5) is as follows:

$$\Delta E = sqrt\left( (x_n - x_s)^2 + (y_n - y_s)^2 + (z_n - z_s)^2 \right) \tag{5}$$

where the HSV spatial coordinates can be calculated as $(x_n, y_n, z_n)$ according to Equation (1), and $(x_s, y_s, z_s)$ is the standard color HSV coordinate.

Figure 9 shows the color deviation of the two materials before and after color calibration in digital photos at different ambient color temperatures. The results indicate that the introduced color calibration methods can significantly reduce the color deviation of digital images when the color temperature is warm or cold.

**Table 4.** Materials, apparatus, and software.

| Materials | | | |
|---|---|---|---|
| **ID** | **Façade Material Name** | **Façade Color Samples** | **Standard HSV Value** |
| MAT. 1 | Ceramic tiles | 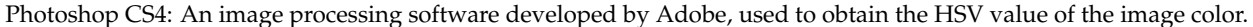 | H:198, S: 8%, V: 96% |
| MAT. 2 | Veneer brick | | H: 16, S: 11%, V: 51% |
| Apparatus/Product | | | |
| Digital camera/Canon EOS 60D | | | |
| Software/Contents | | | |
| Photoshop CS4: An image processing software developed by Adobe, used to obtain the HSV value of the image color. | | | |

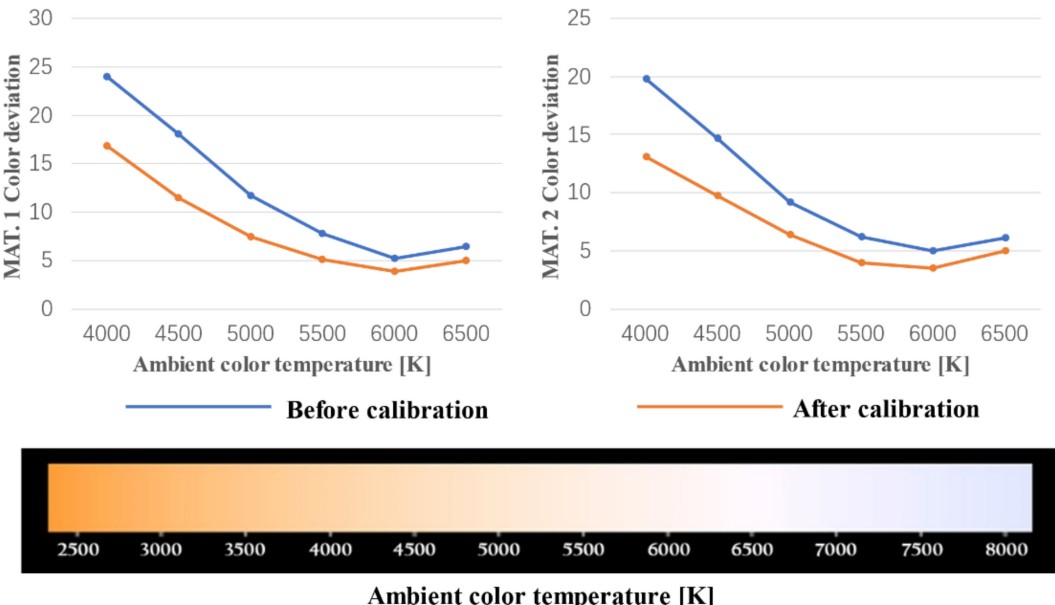

**Figure 9.** The color deviation of the two materials before and after color calibration in digital photos at different color temperatures.

### 4.3.3. Classification Accuracy of Multi-Label Building Functions

As illustrated in Figure 10 and Table 5, the area under the curve (AUC) of the four trained DCNN models was evaluated by our test data. AUC is the area enclosed by the coordinate axis under the receiver operating characteristic (ROC) curve. Since the maximum value of $x$ and $y$ after normalization is 1, and the ROC curve is generally above the line $y = x$, the AUC takes values in the range of 0.5 and 1. The closer the AUC is to 1.0, the higher the authenticity of the detection method. When it is equal to 0.5, the authenticity is the lowest and has no application value [47]. As shown in the results, the overall classification performance of EfficientNet was worse than the other networks. For the accuracy of commercial service and public service classification, ResNeSt performed better than the other three. For the class of residence (R), InceptionNet_v4 achieved the highest AUC value. After comparison, we chose the trained ResNeSt model, which has the highest overall accuracy among the four models, for the following generation of building functional classification maps.

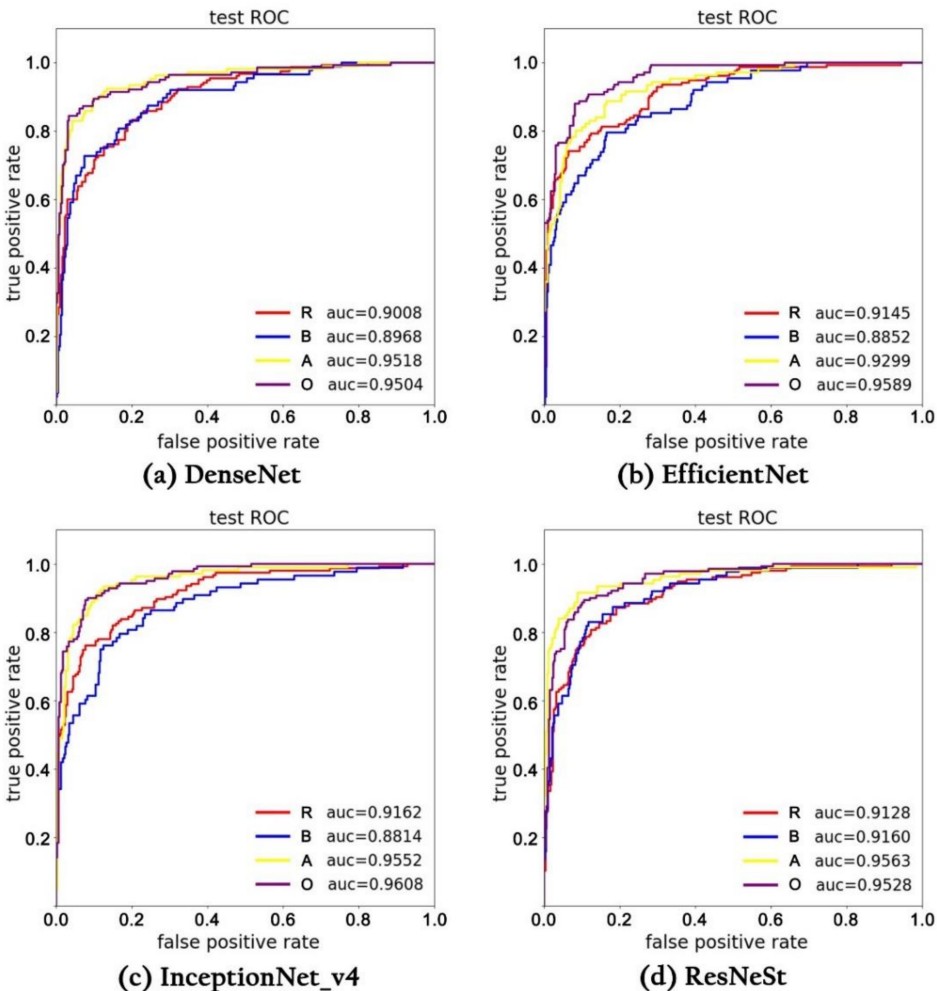

**Figure 10.** The area under the ROC curve (AUC) of all the trained networks evaluated by our test images, including DenseNet (**a**), EfficientNet (**b**), InceptionNet_v4 (**c**), and ResNeSt (**d**). The red line R represents the AUC of residence, the blue line B represents the AUC of commercial service, the yellow line A represents the AUC of public service, and the purple line O represents the AUC of other facilities.

**Table 5.** Multi-label classification performance of all the trained networks.

| Type | DenseNet | EfficientNet | InceptionNet_v4 | ResNeSt |
|---|---|---|---|---|
| Residence (R) | 0.9008 | 0.9145 | **0.9162** | 0.9148 |
| Commercial service (B) | 0.8968 | 0.8852 | 0.8814 | **0.9160** |
| Public service (A) | 0.9518 | 0.9299 | 0.9552 | **0.9563** |
| Other facilities (O) | 0.9504 | 0.9589 | **0.9608** | 0.9528 |
| Overall | 0.9249 | 0.9221 | 0.9284 | **0.9349** |

Bold values represent the highest output achieved among all the listed DCNNs.

### 4.3.4. Validation Based on Field Investigation Data

We further validated the proposed methodology in terms of color measurement and functional classification of building façades based on 200 field survey images of street views randomly extracted from the three study areas. The comparisons between the field survey and our proposed measurement method are shown in Figure 11. For color measurement validation, we first visually compared the architectural standard color card with the surveyed façade and recorded the color code closest to the investigated object as the ground truth. Then, we took pictures of the surveyed building façade and used our method to obtain the HSV value of the measured façade color. Finally, the color deviation

between the measured color and ground truth was calculated for each field survey sample, and the range of color deviation was counted. The histogram of color deviation is shown in Figure 12, and more than 67% of the color deviation is lower than 20. For classification validation, we compared the classification results of building functionality by our approach with the ground truth, and the overall building functional classification accuracy is 86.5%, as shown in Table 6. Most categories exceeded 85% accuracy, except for the residential type. These results are similar to the classification accuracy in Figure 10d and show that the prediction results by the trained ResNeSt achieve consistency with the verification results of the field investigation data.

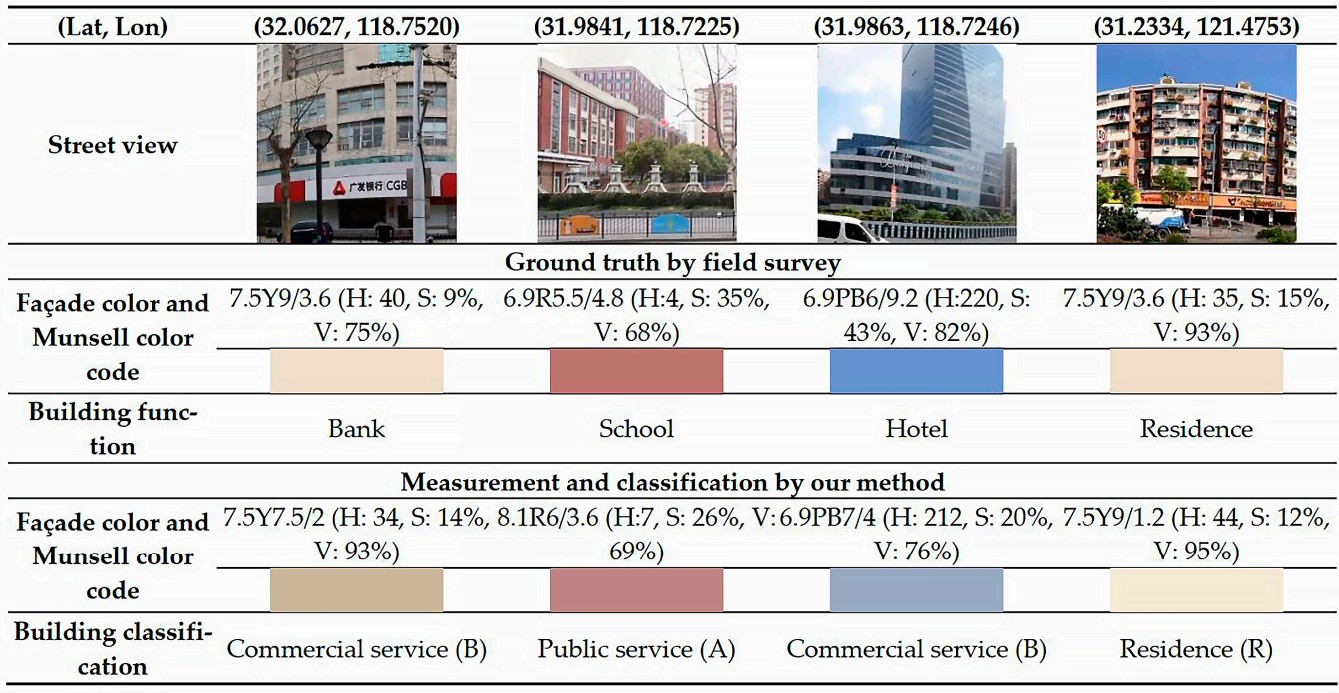

| (Lat, Lon) | (32.0627, 118.7520) | (31.9841, 118.7225) | (31.9863, 118.7246) | (31.2334, 121.4753) |
|---|---|---|---|---|
| Street view | | | | |
| **Ground truth by field survey** | | | | |
| Façade color and Munsell color code | 7.5Y9/3.6 (H: 40, S: 9%, V: 75%) | 6.9R5.5/4.8 (H:4, S: 35%, V: 68%) | 6.9PB6/9.2 (H:220, S: 43%, V: 82%) | 7.5Y9/3.6 (H: 35, S: 15%, V: 93%) |
| Building function | Bank | School | Hotel | Residence |
| **Measurement and classification by our method** | | | | |
| Façade color and Munsell color code | 7.5Y7.5/2 (H: 34, S: 14%, V: 93%) | 8.1R6/3.6 (H:7, S: 26%, V: 69%) | 6.9PB7/4 (H: 212, S: 20%, V: 76%) | 7.5Y9/1.2 (H: 44, S: 12%, V: 95%) |
| Building classification | Commercial service (B) | Public service (A) | Commercial service (B) | Residence (R) |

**Figure 11.** Comparison of our proposed measurement method with the field survey data.

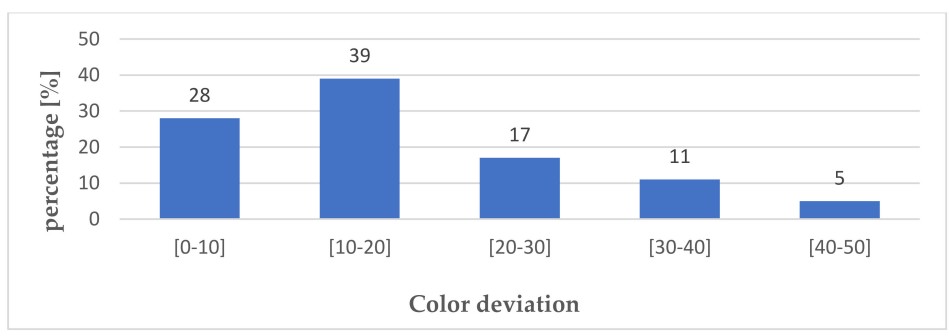

**Figure 12.** Color deviation histogram based on survey data measurement.

**Table 6.** Building classification accuracy for the 200 sampled images.

| Type | R | B | A | O | R + A | B + A |
|---|---|---|---|---|---|---|
| Number of samples | 46 | 42 | 38 | 30 | 20 | 24 |
| Subclass accuracy | 84.8% | 88.1% | 89.5% | 86.7% | 85% | 87.5% |
| Overall accuracy | | | 86.5% | | | |

### 4.3.5. Mapping of the Façade Color and Building Classification in the Three Study Areas

256,010 images with coordinates from the three cities were used to calculate the building façade color while predicting the building function by our proposed method. Table 7 represents the data structure of the statistical results. Figures 13 and 14 show the measurement map of the dominant façade color and the distribution map of building functions in the valid street view images of the study area.

Previous studies have shown that the colors will converge in specific areas when there is a high degree of functional uniformity and the materials chosen for the building are similar [6]. We take three commercial areas in cities as examples, namely Hefei Government Affairs and Culture District (Zone H), Nanjing Hexi Central Business District (Zone X), and Shanghai Lujiazui (Zone L), to demonstrate the color predisposition of buildings based on functional division. Figures 13a and 14a show that the building classification results are mainly for commercial services in Zone H, followed by other facilities and public services. The color calculation results show that this area is predominantly gray (N4.5 to N7.5) and blue (5PB to 5B), consistent with the actual situation. As shown in Figures 13b and 14b, blue is the main façade color in Zone X of Nanjing, concentrated in the blue-violet interval (7.5PB to 5B). In the Zone L area of Shanghai, as shown in Figures 13c and 14c, the calculation results of the building classification show that the building classification is mainly for commercial services. The color calculation results show that the façade color in this area is mainly blue (5PB to 5B) and yellowish red (5Y to 5YR), and commercial buildings cause the blue color, and the yellowish-red color primarily exists on the exterior walls of residences.

### 4.3.6. Statistical Results of Façade Color Corresponding to Building Classification

Table 8 shows the statistical results of the dominant façade color corresponding to the building functional classification in the three cities. In the commercial buildings of the study areas, the color is mainly blue, and the proportion of Shanghai is the largest at 47.3%; Nanjing has the lowest percentage at 31.8%, showing the color of commercial service buildings in Shanghai is highly related to their functions. For the dominant color of public service buildings, the situation is different in the three cities, showing that they have various color styles. In residential buildings, gray and yellow-red occupy the primary type in the three cities, and these results are similar to the site survey study of A. Gou et al. [6,48] Although the buildings with an obvious color predisposition are usually popular with the public, the actual situation is that most buildings have a low chromaticity in urban streets.

**Table 7.** The data structure of façade color measurement and building classification results. The Munsell color system divides hues into ten kinds of colors: red (R), red-purple (R.P.), purple (P), purple-blue (P.B.), blue (B), blue-green (B.G.), green (G), green-yellow (G.Y.), yellow (Y), and yellow-red (Y.R.).

| Picture ID | Dominant Color Measurement | | | | | Building Function | Latitude | Longitude |
|---|---|---|---|---|---|---|---|---|
| | Munsell Color Code | H | S | V | Color Sample | | | |
| 1 | 8.1GY6/1.4 | 192 | 10% | 61% | | A | 32.0212 | 118.7632 |
| 2 | 10YR9/1 | 37 | 10% | 94% | | B | 32.0214 | 118.7636 |
| 3 | 10YR8.5/4 | 24 | 13% | 86% | | R | 32.0605 | 118.7798 |
| 4 | 6.9PB7/4 | 212 | 20% | 76% | | B | 31.9863 | 118.7246 |
| 5 | 8.1R6/3.6 | 7 | 26% | 69% | | A | 31.9841 | 118.7225 |

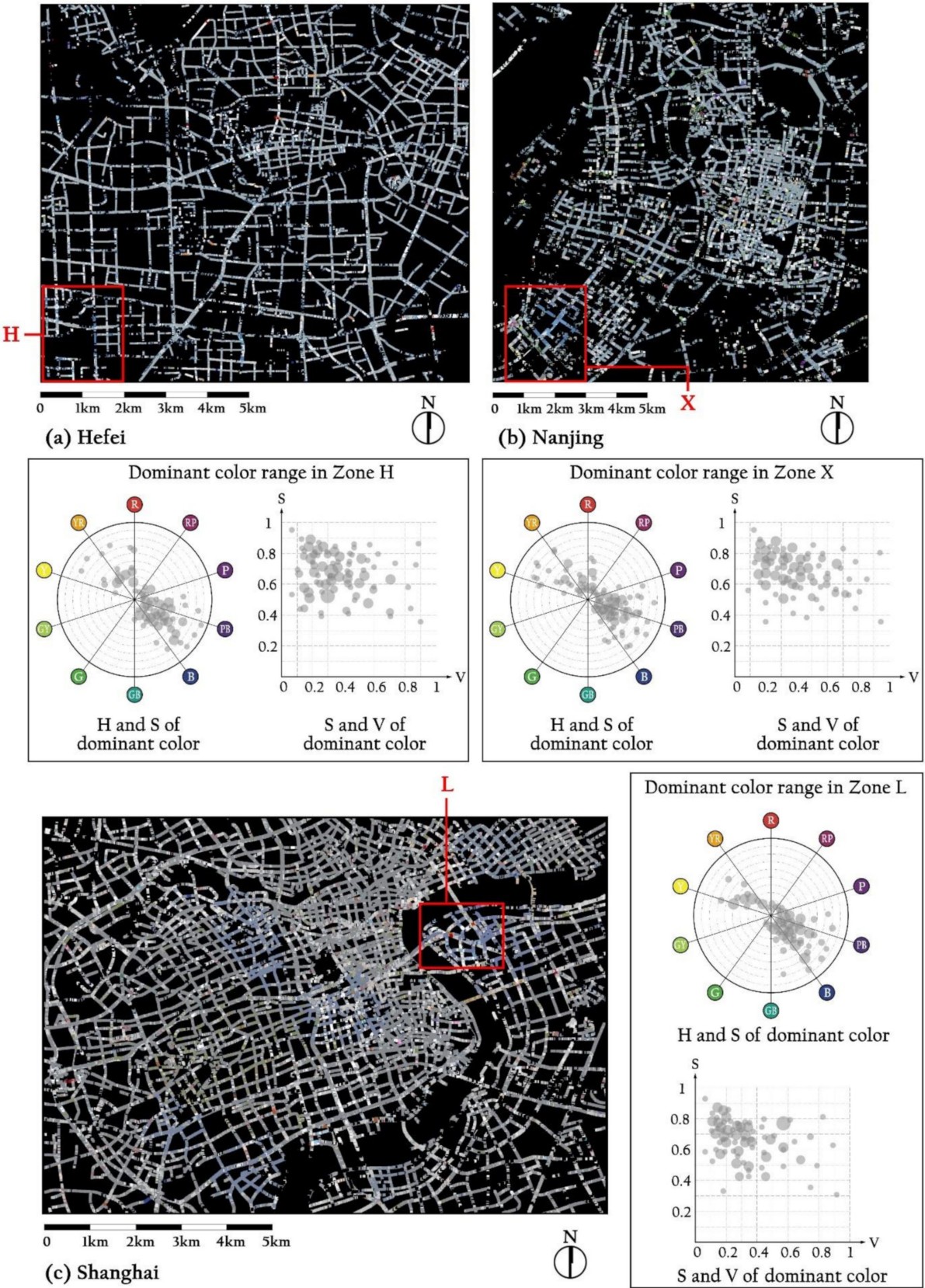

**Figure 13.** Mapping the dominant color of the building façade of Hefei, Nanjing, and Shanghai.

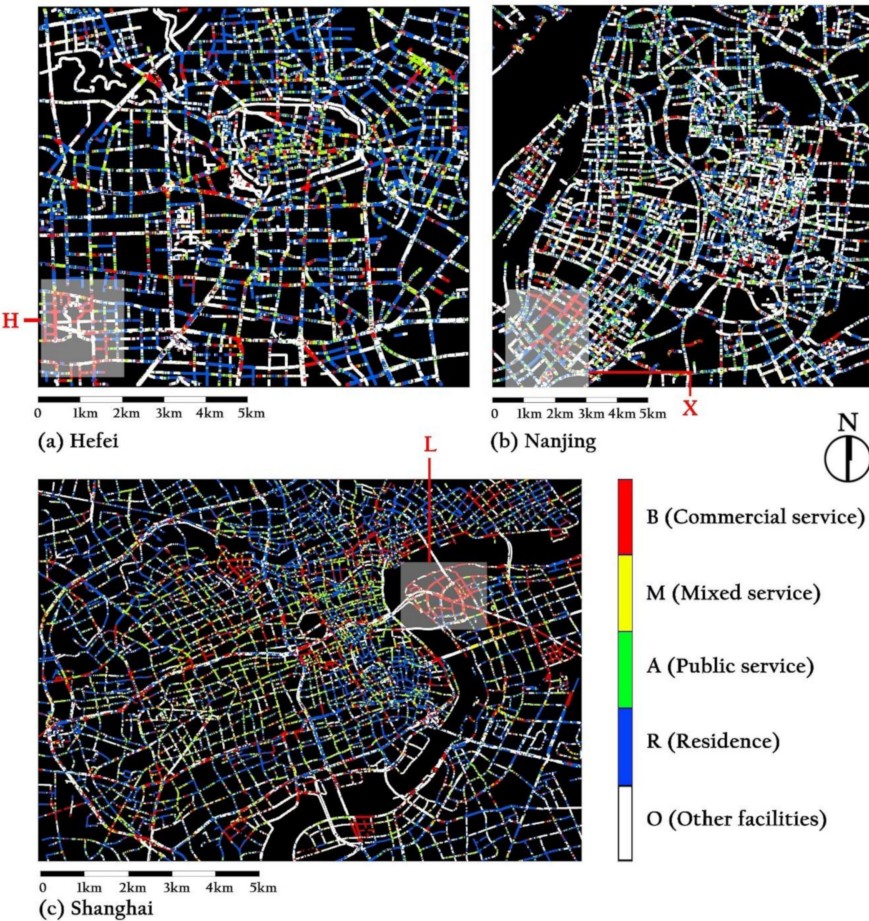

**Figure 14.** Mapping the building function of Hefei, Nanjing, and Shanghai.

**Table 8.** The statistical results of façade color on building classification in the three study areas.

| Hefei | | | | | |
|---|---|---|---|---|---|
| Classification | 1st color | Proportion | 2nd color | Proportion | Proportion of other colors |
| Commerce | Blue | 38.5% | Gray | 32.6% | 28.9% |
| Public service | Gray | 20.6% | Yellow-red | 19.9% | 59.5% |
| Residence | Gray | 42.4% | Yellow-red | 16.1% | 41.5% |
| Other facilities | Gray | 61.2% | White | 13.9% | 24.9% |
| Nanjing | | | | | |
| Classification | 1st color | Proportion | 2nd color | Proportion | Proportion of other colors |
| Commerce | Blue | 31.8% | Bluish-Gray | 20.8% | 47.4% |
| Public service | Bluish-Gray | 20.8% | Bluish-Gray | 18.7% | 60.5% |
| Residence | Gray | 28.2% | Yellow-red | 17.7% | 54.1% |
| Other facilities | Greenish-Gray | 68.5% | Gray | 9.6% | 21.9% |
| Shanghai | | | | | |
| Classification | 1st color | Proportion | 2nd color | Proportion | Proportion of other colors |
| Commerce | Blue | 47.3% | Gray | 9.6% | 43.1% |
| Public service | Warm Gray | 24.6% | Greenish-Gray | 14.8% | 60.6% |
| Residence | Yellow-red | 29.8% | Gray | 18.4% | 51.8% |
| Other facilities | Gray | 51.7% | White | 8.7% | 39.6% |

Moreover, this approach allows not only the analysis of the overall façade color at the city scale but also the identification of details in the specific locations. For example, in the inner city of Shanghai, the hue, saturation, and brightness of the urban street are distributed in groups, and there are apparent urban color characteristics of regional color differences. In some historic districts of Nanjing, the building colors avoid intense colors and resemble the historical heritage, reflecting the respect for culture and history in urban planning. As shown in Table 8, the dominant façade color ratio of each functional area in Hefei is more remarkable than the other two, indicating a convergence of urban colors. This study may help city managers perceive the urban color identity and provide data support for future urban color planning.

## 5. Discussion

This section discusses this work regarding the comparison with conventional methods, potential applications, and limitations of the proposed method.

### 5.1. Comparison with Conventional Methods

In previous manual measurements, the methods developed by Li et al. [9] and Nguyen et al. [10] are computationally expensive in terms of façade color measurement and building function statistics, based mainly on qualitative analysis, and with low expansibility. These methods require a significant amount of manual measurement data, including on-site streetscape images and questionnaires, and are restricted to neighborhood-scale studies. By contrast, our proposed deep learning-based data processing method can analyze the data in large quantities with high accuracy and is more cost-efficient in measuring the façade color corresponding to the building classification than the field survey-based method. The proposed method can quantitatively analyze the color distribution at different building functions to support evidence-based urban analytics and design rather than simply qualitative descriptions. In addition, due to the wide coverage and frequent updates of the street view services, which give sufficient street-view data, our method can be applied to large-scale urban color and function studies in different cities, especially in fast-growing areas without enough time for field surveys.

### 5.2. Potential Applications

This study is a preliminary attempt to construct a quantitative research method for the city-scale measurement of façade color and functions. After testing, the technique demonstrated its viability and convenience in initial investigations of urban design, implying potential application as an augmented tool for designers to establish objective decision bias and enable a data-driven strategy. Given the method's benefits, it could be used to discover discordant architectural colors in particular functional areas, assess the color planning of the built environment, and provide foundation color details for urban design implementation, thus facilitating a feedback process. For example, the new and old façade color has a noticeable difference because of the pace of construction and business distribution. This study provides city managers with a clear understanding of street-level façade colors with building classification to realize the optimal balanced development of the new buildings and traditions. In addition, quantitative measurement and classification provide empirical value for intelligent design guidelines in various areas, such as residential, commercial, and public service. By analyzing the color and function of the city, the authorities could explore the color tendencies of functional buildings in different cities and propose urban planning solutions with their own identity while avoiding the drawbacks of stylistic homogenization induced by the prevalence of functionalism. It is expected to help improve the color quality of the urban built environment, especially in further exploring the visual environment design, to better support urban renewal in the post-urbanization period.

*5.3. Limitations*

The intense sunlight will impact the quality of street view images, affecting the color calculation based on our introduced method; an example is shown in Figure 15a. The color calibration of street view images can improve the accuracy of the calculation results. However, for some overexposed and overly dark street view images, it is difficult to obtain the actual color of the building façade with the currently used color correction methods. Besides, from the building classification results of the four classes, some residential areas are relatively more complicated to identify than other classes since residential areas in older towns tend to be highly mixed in function. Commercial services often exist on the ground floor of residences, and few individual houses are in the streets of these study cities, causing the classification accuracy of some residential buildings to be lower than other classes. As shown in Figure 15b, the building in the street view image is predicted to be a mixed service by the proposed method. Last, there are a few manual tagging errors from OSM users in the training set of the classification model, especially for similar façade featuring. As shown in Figure 15c, the building in the street view photo tends to be a residential apartment, while the label from the OSM user is a hotel.

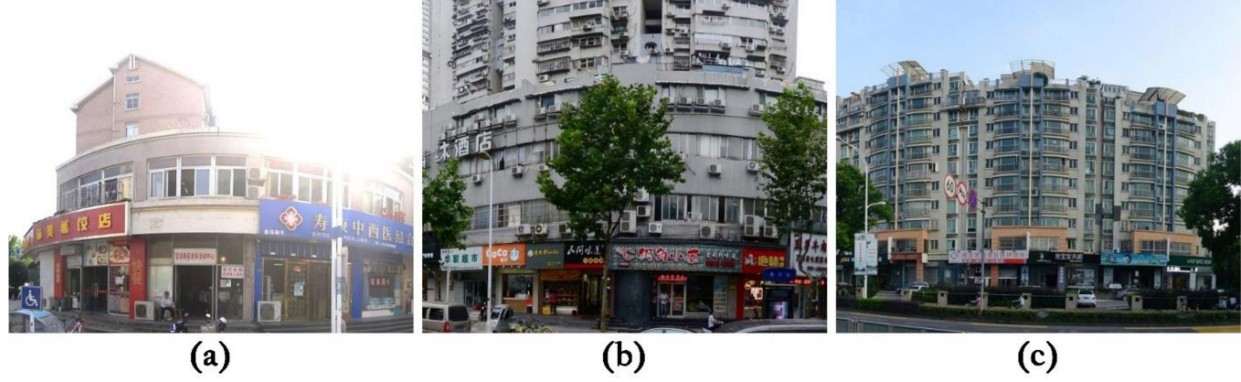

**Figure 15.** Some examples from street view images show the limitations of the proposed method. (**a**) Despite the color calibration, color deviations remain in the overexposed street view image. (**b**) The residential building with commercial service is not easy to identify. (**c**) The building in the street view photo tends to be a residential house, while the label from the OSM user is a hotel.

## 6. Conclusions

In summary, this study proposed an automatic measurement approach for façade color while classifying building functions at the city scale by applying state-of-the-art deep learning methods and street view images. A data pre-processing method for façade color measurement was developed in several steps: image color calibration, building façade segmentation, and dominant color calculation. A benchmark dataset of street view images is built for training a multi-label classifier for building functions, including residential, public services, commercial services, and other facilities.

We applied our method to measure façade color and classify building functions in three cities in China, and the accuracy of the proposed method was verified by field surveys. The results show that our method has satisfactory accuracy, with an IoU accuracy of 87.13% for building façade segmentation, a color deviation of less than 20 for more than 67% of the measured data, and overall accuracy of 86.50% for the building functional classification. Compared to the previous methods, our method overcomes the difficulties of applying manual sample collection on a large scale and enables quantitative analysis of the relationship between building colors and functions. The mapping results of the cities can be applied to urban analytics and urban planning, such as evaluating the urban color identity and providing foundation information for urban renewal.

In future work, it is promising to select high-quality street view images to improve the accuracy of color measurement and building classification since the street view service

can choose photos of the same location on cloudy days. Furthermore, in the training set of deep learning-based image classification, a small number of non-functional buildings can decrease the accuracy of the classifier. By obtaining the exact function of each building from the city geographic information platform, the accuracy of building classification benchmarks can be improved. Finally, the current analysis is mainly aimed at experts rather than non-experts and focuses on the quantitative analysis of urban color and function. As a next step, we plan to build an easy-to-understand visualization platform to promote public participation in urban color planning and build a consensus for the development of a good urban visual environment.

**Author Contributions:** Conceptualization, Jiaxin Zhang and Tomohiro Fukuda; methodology, Jiaxin Zhang, Tomohiro Fukuda and Nobuyoshi Yabuki; data curation, Jiaxin Zhang; writing—original draft preparation, Jiaxin Zhang; writing—review and editing, Jiaxin Zhang, Tomohiro Fukuda and Nobuyoshi Yabuki; funding acquisition, Tomohiro Fukuda. All authors have read and agreed to the published version of the manuscript.

**Funding:** This research was supported by JSPS KAKENHI Grant Number JP19K12681.

**Institutional Review Board Statement:** Not applicable.

**Informed Consent Statement:** Not applicable.

**Data Availability Statement:** Not applicable.

**Acknowledgments:** We would like to thank the editors and anonymous reviewers for their constructive suggestions and comments, which helped improve this paper's quality.

**Conflicts of Interest:** The authors declare no conflict of interest.

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
