# Peer review of "Development of a City-Scale Approach for Façade Color Measurement with Building Functional Classification Using Deep Learning and Street View Images"

_ijgi, doi:10.3390/ijgi10080551_

Round 1
Reviewer 1 Report
The article proposes p a city-scale method for building facade color measurement with building functional classification using deep learning and street-level imagery.
The Abstract can better establish the applications and motivations of the proposed solution.
While the introduction properly connects the proposed approach to the previous work, it does not provide a clear list of contributions and how scientifically the proposed solution can surpass other baselines.
Going straight from the literature review to the framework overview does not seem ideal. I recommend the authors provide a few definitions followed by a clear problem statement before the methodology section. Authors would better use mathematical notations.
In the conclusion section, you need to report the experimental results. Having a bulleted list of application-oriented findings is not ideal to reflect the real contributions.
The work more seems aligned toward the application perspective, rather than research work, where the formulas are more based on heuristic findings, e.g. Eq. 1.
I am unsure if the China Building Color Chart (CBCC) neither affects the proposed solution for the building facade color measurement nor the claim that the method is generalizable.
The proposed solution in this work is mainly applicational and based on heuristics. Using the methods like Multi-label classification, Facade color calculation, and Pyramid Scene Parsing Network (PSPNet) is not considered as novel contributions for the research paper.
The experiment is mainly focused to study the effectiveness of the main proposed method. Authors would better include a section for the baselines and explain/discuss why the proposed solution can excel other state-of-the-art methods. Also, using standard distance for HSV is not the only model to compute the accuracy, what about Precision and Recall where we indicate how many time our methods have failed to detect the building correctly.
Author Response
Dear Reviewer,
We would like to thank you for the positive, valuable, and detailed review of our manuscript. We are convinced that the points you raised contributed significantly to the improvement of the article.
To better explain the changes I made to your comment, I have attached the RESPONSE in the attachment below, please download it, and thank you once more.
Yours sincerely,
Tomohiro Fukuda
Corresponding author

Reviewer 2 Report
The paper is clear and well written.
Author Response
Dear Reviewer,
We would like to thank you for the positive, valuable, and detailed review of our manuscript entitled "Development of a city-scale approach for facade color measurement with building functional classification using deep learning and street view images" (Manuscript ID: ijgi-1239304). We appreciated for reviewers' warm work earnestly.
Once again, thank you very much for your comments.
Yours sincerely,
Tomohiro Fukuda
Corresponding author
Reviewer 2 Comment
The paper is clear and well-written.
Authors' Response
Thank you for your positive comments.
Reviewer 3 Report
The paper describes an implementation of deep learning approaches to classify buildings by their functions and detect the facade colours. Furthermore, an analysis of the relationship between colour and functions is carried out.
It is a very interesting work, with great potential for future applications in planning. The methods are sound and well described as well as the results and their discussion.
However, the introduction and literature review can be much improved. In many parts it is not clear what the authors want to say (i.e. what does the following sentence mean? The authorities conducting urban color research can continue the urban context, respect historical heritage, and guide new construction development).
I would strongly suggest a major revision of the first two sections before publishing.
Author Response
Dear Reviewer,
We would like to thank you for the positive and constructive feedback of our manuscript entitled "Development of a city-scale approach for facade color measurement with building functional classification using deep learning and street view images" (Manuscript ID: ijgi-1239304). We are convinced that the points you raised are valuable for improving the work.
We have taken advantage of the reviewing period to address those comments point-by-point (Please see the attachment and download it). We hope you find our response and the appropriate modifications of the manuscript we attach behind the response file.
Best regards and thank you once more.
Yours sincerely,
Tomohiro Fukuda
Corresponding author
